# Enterocin 7420 and Sage in Rabbit Diet and Their Effect on Meat Mineral Content and Physico-Chemical Properties

**DOI:** 10.3390/microorganisms10061094

**Published:** 2022-05-25

**Authors:** Monika Pogány Simonová, Ľubica Chrastinová, Andrea Lauková

**Affiliations:** 1Centre of Biosciences of the Slovak Academy of Sciences, Institute of Animal Physiology, Šoltésovej 4-6, 04001 Košice, Slovakia; laukova@saske.sk; 2Institute for Nutrition, National Agricultural and Food Centre, Hlohovecká 2, 95141 Lužianky, Slovakia; lubica.chrastinova@nppc.sk

**Keywords:** enterocin, sage extract, feed additives, mineral profile, rabbit meat

## Abstract

Rabbit meat has outstanding nutritional characteristics—it is a lean meat with low fat, cholesterol and sodium content, with high-biological-value proteins, potassium, phosphorus, selenium, iron and vitamin B12 level. The dietary inclusion of natural bioactive compounds can improve the quality of rabbit meat. The present study evaluated the effect of enterocin 7420 (Ent 7420) and sage (*Salvia officinalis*) extract on the quality and mineral content of rabbit meat. A total of 96 Hyla rabbits (aged 35 days) were divided into E (Ent 7420; 50 µL/animal/d), S (sage extract; 10 µL/animal/d), E + S (Ent 7420 and sage in combination) and control (C) groups. Additives were administrated in drinking water for 21 days. A significant increase in meat iron (*p* < 0.01) content was noted; phosphorus and zinc levels were also elevated in experimental groups, compared with control data. Ent 7420 and sage treatment reduced the calcium and manganese (*p* < 0.01) contents. The physico-chemical traits of rabbit meat were not negatively influenced by treatment. Based on these results, diet supplementation, mostly with Ent 7420 but also in combination with sage, could enhance the quality of rabbit meat mineral, with a focus on its iron, phosphorus and zinc contents.

## 1. Introduction

Rabbit meat is an excellent source of minerals and trace elements, such as potassium, calcium, phosphorus, and selenium and has the highest concentration of iron in any type of meat. It is rich in vitamins, mainly vitamin B3, B6, B12 and E, and in Omega-3 and six fatty acids. Another advantage of rabbit meat is its low sodium level. For this reason, it is recommended for children, pregnant women and people with high blood pressure. It also contains easily digestible proteins, with low amounts of cholesterol and fat. Even though rabbit meat naturally offers a remarkable nutritious quality, the dietary fortification of rabbits with bioactive compounds has been an increasing trend in recent years, and rabbit meat is becoming a functional food with its superior nutritional properties [1]. Amongst natural feed additives, bacteriocins come to the forefront, not only as commonly used starter cultures and preservatives in food industry, but also in the agriculture sector and veterinary medicine to improve the animals’ health and productivity [2,3,4]. Bacteriocins, antimicrobial substances produced mostly by lactic acid bacteria (LAB; [5]), are usually used in animals to enhance their health status and productivity, due to stabilized intestinal microbiota and mucosal immunity. Mostly, colicins, microcins, lacticin, garvicin and lantibiotic nisin are used in aquacultures, ruminants, poultry and swine production [3,6,7]. Enterocins (bacteriocins produced mostly by enterococci) also have a great antimicrobial and immuno-stimulatory potential but until recently, mostly enterocin-producing strains were applied to piglets and poultry [8,9,10]. Only enterocin (Ent) A (produced by the *Enterococcus faecium* EK13/CCM7419 strain) was in vivo tested in Japanese quails [10]. Rabbits are also a significant part of animal food production, and mostly probiotics and herbal extracts are studied as potential feed additives in their nutrition [11,12,13,14]. To extend the knowledge regarding bacteriocin applications in rabbit farms, nisin, gallidermin, and enterocins 4231, 7420, EF55, A/P and M were supplemented to the rabbits’ diet alone or in combination with phyto-additives [15,16,17,18,19,20,21,22,23,24]. Most of these studies present the bacteriocins/enterocins’ effect on growth performance, intestinal microbial composition and enzymatic activity, gut morphology and the immune response of rabbits. The monitoring of changes in rabbit meat properties due to bacteriocins/enterocins applications is further limited [25,26,27,28,29]. Plants (whole plants, leaves, seeds as feedstuff) and their extracts (applied as additives) are often used in rabbit nutrition due to their ability to stimulate appetite, digestion, immunity and physiological processes, as well as their strong antimicrobial, anti-inflammatory and antioxidant effects. Among herbal extracts and phyto-additives, fennel, thyme, rosemary, sage, and oregano leaves, seeds and extracts are the most often supplemented to diets of rabbit meat, and meat products are enriched with them. There is also a growing interest in sage (*Salvia officinalis*) plants, seeds and extracts for use in animal feeding due to their oil content, which is a source of polyunsaturated fatty acid (PUFA-linoleic and α-linolenic acid). Dietary administration with sage and its extracts/by-products could increase the PUFA content of animal products (eggs, meat [14]). Several previous studies demonstrated that a combined application of enterocins and sage extract did not have a negative influence on the characteristics of rabbit carcasses [25,27]. Moving forward from these results, the objectives of this in vivo study were to determine the effects of non-commercial Ent 7420 and sage extract administration in drinking water, both separately and in combination, on the physico-chemical parameters and mineral composition of rabbit meat.

## 2. Materials and Methods

### 2.1. Animals, Experiment Design and Diet

The experiment was performed in cooperation with our colleagues at the National Agricultural and Food Centre (NAFC, Lužianky-Nitra, Slovakia). All applicable international, national and/or institutional guidelines for animal care were followed appropriately, and the experimental protocol was approved by the Institutional Ethic Committee, and the State Veterinary and Food Administration of the Slovak Republic (permission code: SK CH 17016 and SK U 18016).

A total of 96 weaned Hyla breed male rabbits, aged 35 days (average live weight 767.2 g ± 17.5) were divided into four groups (*n* = 24), each consisting of 6 replicates (1 replicate/4 rabbits/2 cages, 1 cage/2 animals). Rabbits were housed in standard cages (61 cm × 34 cm × 33 cm) in a closed building equipped with heating and a forced ventilation system, which allowed the environmental temperature to be adjusted within the range of 20 ± 4 °C and to a relative humidity (70 ± 5%). The photoperiod was 16L:8D. The animals were fed with a commercial pelleted basal diet for growing rabbits (Table 1) with access to feed and water *ad libitum* during the experiment.

The rabbits in group E were administered enterocin Ent 7420 (a dose of 50 µL/animal/day, with activity of 25,600 AU/mL, from day 0/1 to day 21) in their drinking water, through nipple drinkers. The semi-purified Ent 7420 was prepared according to Simonová and Lauková [29], and its activity was tested using the agar spot test according to De Vuyst et al. [31] against the principal indicator strain *E. avium* EA5 (isolated from piglet in our laboratory) and expressed in arbitrary units per mL (AU/mL). The rabbits in group S received sage plant extract (*Salvia officinalis* extract containing of 24% thujone, 18% borneol, 15% cineole; Calendula, Nová Ľubovňa, Slovakia) in their drinking water at a dose of 10 μL/animal/day. The animals in the E + S groups were administered the combination of Ent 7420 and sage extract. Based on our previous experiments, showing that these additives could be dissolved in distilled water [31], the additives were first applied to 100 mL of drinking water in all cages, and after consuming this volume, the rabbits had access to water ad libitum. Control rabbits (group C) had the same conditions, but without additives being applied to their drinking water, and they were fed a commercial diet. The experiment lasted for 42 days.

### 2.2. Slaughtering, Physico-Chemical and Mineral Analysis

At days 21 and 42, 6 rabbits from each group (*n* = 6, 1 rabbit/1 replicate) were selected based on daily weight measurement to ensure similar weight of animals (day 21; average live weight: 1697.5 g ± 123.5; day 42; average live weight: 2595.2 g ± 169.8). Rabbits were slaughtered after electro-stunning (50 Hz, 0.3 A/rabbit/4 s) in an experimental slaughterhouse by cutting the carotid and jugular veins, and they quickly bled out. *Longissimus thoracis et lumborum* (*LTL*) was separated by removing the skin, fat and connective tissue, before being chilled and stored 24 h at 4 °C until physicochemical and mineral content analysis started.

The ultimate pH was determined at 24 h post mortem using a Radelkis OP-109 measuring device (Jenway, Felsted, UK) with a combined electrode penetrating 3 mm into samples. Color measurements were taken on the *LTL* surface of the carcass at 24 h after bleeding. Color characteristics were expressed using the CIE L*a*b system (lightness—L*, 0: black and 100: white), (redness and greenness—a*; yellowness and blueness—b*) with a Lab Miniscan (HunterLab, Reston, VA, USA) according to the CIE Lab standards. Lightness measurements at room temperature were also taken. Total water, protein and fat contents were estimated using a FoodScaneTM-Meat Analyser (FOSS Analytical, Hilleored, Denmark) by an FT IR method (Fourier Transform infrared Spectroscopy); expressed in g/100. From these values, the energy value was calculated (EC (kJ/100 g) = 16.75 × Protein content (g/100 g) + 37.68 × Fat content (g/100 g)); Strmiska et al. [32]. Water-holding capacity (WHC) was determined by compress method at constant pressure [33]. The analyzed samples (0.3 g in weight) were placed on filter papers (Schleicher and Shuell No. 2040B, Dassel, Germany) with previously weighed tweezers. Together with the papers, samples were sandwiched between Plexiglas plates and then subjected to a pressure of 5 kg for 5 min. The results were calculated from the difference in weight between the slips with the aspirating spot and the pure filter paper. The ash content was determined by mineralization of the samples at 550 °C according to STN 570185.

Macro and micro element analysis samples were ashed at 550 °C, and the ash was dissolved in 10 mL of HCL (1:3). Minerals were determined by AAS iCE 3000 (Thermo Fisher Scientific, Waltham, MA, USA). The phosphorus content was determined by the molybdovanadate reagent on Camspec M501 (Spectronic Campes Ltd., Leeds, UK).

### 2.3. Statistical Analysis

Treatment effects on the meat parameters were analyzed using a two-way analysis of variance (ANOVA), followed by a Bonferroni post hoc test for pair-wise comparisons, where appropriate. Fixed effects for the model included period and treatment and the interaction between them. Random terms included cage. The statistical model included the effects of period and treatment and their interactions. Data are expressed as means and standard deviations (SD). Mean values within the same row with different superscripts indicate a significant difference *p* ≤ 0.05. All statistical analyses were performed using GraphPad Prism statistical software (GraphPad Prism version 6.0, GraphPad Software, San Diego, CA, USA).

## 3. Results

The physico-chemical characteristics of the *LTL* are shown in Table 2. Only the time effect was noted on the meat energy value. No negative effect of tested additives was noted on the analyzed parameters. Reduced levels (although not significant) of pH24, WHC and energy values were found in all experimental groups compared to control data, while water content slightly increased.

Ent 7420 and sage treatment reduced calcium (day 21; E vs. E + S: *p* < 0.05; E, S vs. C: *p* < 0.01; day 42; S vs. E, E + S, C: *p* < 0.001; E + S vs. S, C: *p* < 0.001) and manganese (day 21; E + S vs. E, S, C: *p* < 0.01; E, S vs. C: *p* < 0.01; day 42; E, S, E + S vs. C: *p* < 0.01) contents and significantly elevated iron levels (day 21; E vs. S: *p* < 0.001; E vs. E + S, C: *p* < 0.01; Table 3). While sage application mostly influenced the macro minerals, with the highest phosphorus (S), magnesium and natrium (E + S) levels, Ent 7420 (E) application elevated potassium levels, while iron and zinc and reduced the calcium and copper concentrations, demonstrating the highest/lowest values.

## 4. Discussion

The Ent 7420 and sage application did not negatively influence the physico-chemical properties of rabbit meat, similarly to previous results achieved through bioactive compounds, such as the supplementation of bacteriocins, herbal extracts and beneficial strains to rabbits [26,27,28,34,35,36]. Meineri et al. [37] and Rotolo et al. [38] also reported the adverse effects of chia seeds and dried leaves on the traits of rabbit meat quality. However, no significant changes within the tested parameters were determined. The pH and WHC decreased with increasing age, similar to findings presented by Pogány Simonová et al. [34,35] and Koziol et al. [39]. The pH (acidity) of rabbit meat is an essential parameter, indicating its shelf life and preventing the microbial growth (bacteriostatic effect of low pH), technological usability and quality of the rabbit meat, which also depends on many factors, such as stress during transport and slaughter, the extent of debleeding and muscle type. Although higher pH values of LTL samples were found when compared with our previous studies, they were still under or at the upper limit of bibliographic values [40,41] and did not negatively influence the positive quality of meat. There is a relation between pH and WHC (increase in pH, increase in WHC); this finding was confirmed during the beneficial *E. faecium* CCM7420 and EF9a strains administration to rabbits [35,36], while enterocins and sage extract supplementation [27,28,34] showed the opposite effect (decreased pH, increased WHC), as was also found in the recent experiment with Ent7420 and sage addition. Lower energy values (but within the range of bibliographic values; [1,40]) of rabbit meat were found compared to other enterocins and sage extract applications [27,28,34], but these values were still higher than those after beneficial *E. faecium* strain supplementation [35,36].

Rabbit meat mineral content has a great variability. Most of tested minerals (except calcium) were measured at lower levels than previously presented by Dalle Zotte and Szendrő [1], Hermida et al. [42], and Nistor et al. [43]. Going forward from the increased levels of most tested minerals in experimental groups, we hypothesize an enhanced nutrient uptake from the intestine, and better mineral inclusion in rabbit meat. It was also interesting to find out that the combined administration of Ent 7420 and sage affected some minerals, such as phosphorus (S), iron, zinc (E), calcium and copper (E, S) in the opposite way when compared to their separate applications. These findings suggest a more antagonistic effect of tested compounds in the case of phosphorus (increased in E + S compared to lower levels in E and decreased compared to higher level in S), iron and zinc (reduced in E + S compared to elevated concentration in E and increased compared to lower S level). Regarding the copper value, we assumed the synergistic effect of sage and Ent 7420 (higher level in E + S compared to E, S and C). Nevertheless, further experiments are needed to determine and/or confirm in more detail if there is any other synergistic or antagonistic effect of both additives on the tested minerals. It is known that probiotics and prebiotics can affect intestinal mineral absorption by releasing bone-modulating factors such as phytoestrogens from food [44]. This finding was also noted in this study regarding the bone minerals, but mostly in the highest calcium level during the combination of Ent 7240 and sage extract in rabbits which confirmed the supportive effect of Ent 7420 on higher phytoestrogens release from sage enriched feed due to improved intestinal microbial environment. However, this synergistic interaction of tested bioactive compounds on meat calcium level was noted in E + S. The reduced Ca content compared to control Ca value did not confirm the positive effects of phyto-estrogenic compounds in sage on Ca intestinal absorption (through estrogen receptors within intestinal cells) and/or remodulation of serum Ca levels, as was previously recorded by Pogány Simonová et al. [26]. Probiotics can increase mineral solubility via an increased bacterial production of short-chain fatty acids (SCFA), enlarge the absorption surface by promoting proliferation of enterocytes mediated by bacterial fermentation products, improve gut health, and increase the expression of calcium-binding proteins, mostly elevating calcium and magnesium absorption [44]. Another way to enhance mineral absorption due to ionization and passive diffusion, is the acidic environment as a result of a higher lactic acid formation [45]. Beneficial gut bacteria, also enhanced/optimized by natural feed additives may improve the availability and absorption of polyvalent cations, such as calcium, phosphorus, magnesium, zinc and iron due to optimum pH conditions for enzymatic phytate degradation and reduction. This finding was repeatedly confirmed after *Enterococcus faecium* CCM7420 and CCM8558 probiotic strains dietary inclusion in rabbits, when phosphorus, iron and zinc concentrations in meat were increased compared to untreated animals [35,46]. Bacteriocins in meaning postbiotics (metabolites of beneficial bacteria) can balance/improve the host microbiome in favor of lactic acid bacteria because of their antimicrobial activity, which inhibits the growth of enteropathogenic bacteria, and thus ensures a higher lactic acid production. This may be another explanation of higher iron and zinc intestinal absorption and the inclusion of rabbit meat, mostly as a result of the higher iron level in meat samples from rabbits receiving Ent 7420. This hypothesis is also confirmed by the results showing an increase in phosphorus, iron and zinc after enterocin M application to rabbits [46].

In the case of iron content in food, it is important to know the respective amounts of ferrous (heme; Fe^2+^) iron, sources from meat, liver and meat products, and ferric (non-heme; Fe^3+^) iron, source from legumes, cereals, vegetables and fruits, with a dietary transformation between these two states. Heme iron is the most bioavailable form of iron, due to its coordination with a porphyrin ring hidden inside a globular protein, and this arrangement protects iron from oxidation and insoluble precipitates forming in the intestine, which promote its bioavailability [47]. Although there are only a few studies concerning the probiotic and postbiotic/bacteriocin effect on rabbit meat, the achieved results show an improvement of rabbit meat quality due to its higher iron content. Contrary to us, Shah et al. [48] observed decreased iron and zinc content in rabbit hind leg samples after microbial fermented feed utilization. On the other hand, these authors noted lower copper and manganese levels, similar to our present and previous results [27,35,46]. Diet supplementation with sage extract/chia seeds was shown to be effective in improving rabbit meat nutritional quality, focusing on its fatty acid profile, and this meat can be considered as a functional food [1]. Because there are only a few studies regarding the effect of sage extract on rabbit meat minerals, we can only assume the activity of phenolic compounds in increasing bone mineral content and bioavailability for iron [49]. Further research will be necessary to clarify the effect of sage bioactive components on minerals absorption and its inclusion in meat.

## 5. Conclusions

It seems that diet supplementation with Ent 7420 and sage extract was effective in improving the meat mineral profile. While Ent 7420 significantly elevated the iron content and increased zinc and potassium levels, sage extract beneficially influenced the phosphorus and zinc concentrations of rabbit meat. Reduced calcium and manganese levels were found after Ent 7420 and sage application. Both additives in combination also increased phosphorus, iron, zinc and copper concentrations, without any adverse effects on the physico-chemical properties of rabbit meat. We conclude that diet supplementation, mainly with Ent 7420, can enhance the nutritional quality of rabbit meat.

## Figures and Tables

**Table 1 microorganisms-10-01094-t001:** Composition and ingredients of the basal diet.

Feed Ingredients (%)		Chemical Composition, Minerals and Vitamins (g/kg Feed)	
Dehydrated lucerne meal	36.0	Crude protein (N*6.25)	175.0
Extracted sunflower meal	5.5	Crude fiber	188.3
Oats	13.0	Fat	32.0
Wheat bran	9.0	Ash	66.40
Dry malting sprouts	15.0	Organic matter	847.5
Extracted rapeseed meal	5.5	Acid detergent fiber (ADF)	185.1
Barley	8.0	Neutral detergent fiber (NDF)	315.5
DDGS	5.0	Lignine	42.3
Sodium chloride	0.3	Hemicellulose	148.5
Premix minerals ^1^	1.7	Cellulose	148.8
Limestone	1.0	Starch	127.2
		Calcium	7.5
		Phosphorus	5.9
		Metabolic energy (MJ/kg)	10.3

Abbreviations: DDGS, dried distilled grains with solubles. ^1^ Premix contains per kg: Calcium 6.73 g; phosphorous 4.13 g; magnesium 1.90 g; sodium 1.36 g; potassium 11.21 g; iron 0.36 g; zinc 0.13 g; copper 0.03 g; and selenium 0.2 mg. Vitamin mixture provided per kg of diet: Vitamin A 1,500,000 IU; Vitamin D3 125,000 IU; Vitamin E 5000 mg; Vitamin B1 100 mg; Vitamin B2 500 mg; Vitamin B6 200 mg; Vitamin B12 0.01 mg; Vitamin K3 0.5 mg; biotin 10 mg; folic acid 25 mg; nicotinic acid 4000 mg; and choline chloride 100,000 mg. The metabolizable energy content was calculated using the equation of Wiseman et al. [30].

**Table 2 microorganisms-10-01094-t002:** The effect of Ent 7420 (E), sage extract (S) and their combinative (E + S) application on the meat physico-chemical characteristics of rabbits *Longissimus thoracis* and *lumborum* (*LTL*; mean ± SD).

Parameter	Day of Experiment	E	S	E + S	C	Significance of Effects		
						Treatment	Time	Interaction
pH 24 h after killing	21	5.82 ± 0.06	5.88 ± 0.01	5.84 ± 0.09	5.90 ± 0.01	1.0000	0.9741	1.0000
	42	5.66 ± 0.06	5.67 ± 0.08	5.67 ± 0.09	5.73 ± 0.02			
Water content (g/100 g)	21	75.17 ± 0.11	75.03 ± 0.06	75.20 ± 0.26	74.97 ± 0.47	1.0000	0.9503	1.0000
	42	75.30 ± 0.10	75.63 ± 0.35	75.33 ± 0.29	75.47 ± 0.38			
Protein content (g/100 g)	21	22.50 ± 0.00	22.63 ± 0.06	22.33 ± 0.25	22.63 ± 0.38	1.0000	0.9689	0.9999
	42	22.27 ± 0.06	21.97 ± 0.15	22.60 ± 0.30	22.40 ± 0.37			
Fat content (g/100 g)	21	1.33 ± 0.12	1.33 ± 0.06	1.47 ± 0.21	1.40 ± 0.17	1.0000	0.9854	1.0000
	42	1.43 ± 0.15	1.40 ± 0.20	1.07 ± 0.15	1.23 ± 0.06			
Ash content (g/100 g)	21	1.00 ± 0.00	1.00 ± 0.00	1.00 ± 0.00	1.00 ± 0.00	1.0000	1.0000	1.0000
	42	1.00 ± 0.00	1.00 ± 0.00	1.00 ± 0.00	1.00 ± 0.00			
L* (lightness)	21	50.06 ± 3.30	47.63 ± 2.39	50.48 ± 2.39	49.97 ± 5.46	0.9996	0.9985	0.9983
	42	48.65 ± 5.75	51.42 ± 1.31	46.06 ± 5.74	51.88 ± 3.24			
a* (redness)	21	0.72 ± 0.60	2.82 ± 1.57	3.63 ± 0.35	1.18 ± 0.97	0.4560	0.0809	0.0693
	42	1.64 ± 0.86	0.55 ± 0.20	0.93 ± 0.43	1.30 ± 0.40			
b* (yellowness)	21	7.39 ± 2.23	7.80 ± 1.70	6.86 ± 1.14	7.25 ± 0.91	0.4954	0.4411	0.9302
	42	7.27 ± 0.12	7.26 ± 0.38	5.80 ± 1.09	7.14 ± 0.94			
Water-holding capacity (g/100 g)	21	36.44 ± 0.74	36.91 ± 1.81	37.27 ± 1.91	34.47 ±4.21	0.4967	0.5436	0.2302
	42	35.86 ± 1.56	37.08 ± 1.07	34.04 ± 1.93	36.09 ± 3.50			
Energy value (kJ/100 g)	21	427.12 ± 4.35	429.35 ± 1.89	428.89 ± 8.10	431.86 ± 10.70	0.8339	**0.0060**	0.3941
	42	426.97 ± 4.86	420.69 ± 10.06	418.74 ± 6.28	420.00 ± 7.14			

**Table 3 microorganisms-10-01094-t003:** The effect of Ent 7420 (E), sage extract (S) and their combinative (E + S) application on the meat mineral content of rabbits *Longissimus thoracis* and *lumborum* (*LTL*; mean ± SD).

Parameter	Day of Experiment	E	S	E + S	C	Significance of Effects		
						Treatment	Time	Interaction
Calcium (mg/100 g)	21	6.20 ± 0.01 ^a^	8.20 ± 0.04 ^ab^	11.30 ± 0.04 ^bc^	12.80 ± 0.01 ^c^	**<0.0001**	**<0.0001**	**<0.0001**
	42	8.13 ± 0.01 ^a^	9.10 ± 0.01 ^b^	7.73 ± 0.01 ^ac^	16.77 ± 0.04 ^d^			
Phosphorus (mg/100 g)	21	201.93 ± 0.44	228.47 ± 0.12	218.77 ± 0.06	206.20 ± 0.18	**0.0126**	0.8122	0.3519
	42	221.03 ± 0.17 ^a^	185.63 ± 0.26 ^b^	195.33 ± 0.04 ^ab^	225.33 ± 0.09 ^b^			
Magnesium (mg/100 g)	21	25.10 ± 0.01	25.37 ± 0.01	25.40 ± 0.01	24.77 ± 0.01	0.6447	0.1187	<0.0001
	42	26.73 ± 0.01	26.30 ± 0.01	27.07 ± 0.01	26.13 ± 0.01			
Natrium (mg/100 g)	21	31.63 ± 0.02	30.53 ± 0.02	34.10 ± 0.03	29.27 ± 0.02	1.0000	1.0000	0.9684
	42	29.07 ± 0.02	29.73 ± 0.01	25.17 ± 0.01	29.50 ± 0.01			
Potassium (mg/100 g)	21	413.67 ± 0.14	400.47 ± 0.03	396.67 ± 0.06	401.77 ± 0.22	0.6994	0.3619	0.1928
	42	411.53 ± 0.16	410.73 ± 0.10	407.43 ± 0.03	406.30 ± 0.11			
Iron (mg/100 g)	21	0.579 ± 0.035 ^a^	0.341 ± 0.124 ^b^	0.465 ± 0.092 ^b^	0.365 ± 0.146 ^b^	**0.0491**	**0.0088**	0.2693
	42	0.481 ± 0.035	0.410 ± 0.081	0.355 ± 0.024	0.465 ± 0.053			
Manganese (mg/100 g)	21	0.064 ± 0.004 ^a^	0.061 ± 0.009 ^a^	0.066 ± 0.025 ^b^	0.084 ± 0.010 ^a^	0.0775	**<0.0001**	0.0524
	42	0.029 ± 0.012 ^a^	0.027 ± 0.051 ^a^	0.018 ± 0.007 ^a^	0.082 ± 0.048 ^b^			
Zinc (mg/100 g)	21	1.350 ± 0.244	1.123 ± 0.141	1.216 ± 0.143	1.043 ± 0.190	0.2357	**0.0126**	**0.0002**
	42	1.779 ± 0.517 ^a^	1.647 ± 0.504 ^a^	1.989 ± 0.485 ^ab^	1.190 ± 0.131 ^b^			
Copper (mg/100 g)	21	0.117 ± 0.017 ^a^	0.119 ± 0.015 ^a^	0.195 ± 0.025 ^b^	0.120 ± 0.079 ^a^	**0.0026**	0.0519	0.0824
	42	0.208 ± 0.034 ^a^	0.109 ± 0.001 ^b^	0.138 ± 0.013 ^bc^	0.200 ± 0.066 ^ac^			

^a^, ^b^, ^c^–mean values marked with different letters differ significantly at *p* ≤ 0.05.

## Data Availability

Data are available upon reasonable request to the corresponding author.

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
