# Peer review of "Enterocin 7420 and Sage in Rabbit Diet and Their Effect on Meat Mineral Content and Physico-Chemical Properties"

_microorganisms, 2022, doi:10.3390/microorganisms10061094_

Round 1
Reviewer 1 Report
GENERAL COMMENT:
I consider this work is within the scope of “Microorganims”. It contains information useful in a field in which available information is of interest to improve knowledge on rabbit feeding and nutrition and meat quality. Overall, it is well organised. However, I indicate several flaws found in the manuscript. I indicate these flaws below and in a commented PDF file I have uploaded.
ABSTRACT:
Line 14: Add Latin name of sage (Salvia officinalis) the first time it appears in the Abstract.
INTRODUCTION:
Add Latin name of sage (Salvia officinalis) the first time it appears in the Introduction.
Lines 36-37: Broken line; correct it.
MATERIALS AND METHODS:
Line 78: If available, add average live weight of rabbits at weaning.
Line 93: Insert: "were" where indicated.
Line 101: Insert: "were" where indicated.
Line 102: Insert: "showing" where indicated.
Lines 109-110: Insert average weight of rabbits at days 21 and 42 of the fattening period.
Line 119: Add bibliographic citation and reference for colour measurement method CIE lab.
Line 135: Please check whether the correct is "HCl" rather than “HCL”.
Line 143: "pen" or "cage"?
Lines 148-155: remove this text.
RESULTS AND DISCUSSION:
These sections are correctly arranged.
CONCLUSIONS:
Conclusions are correct.
REFERENCES SECTION:
In general terms, this section is well organised and adjusted to the style of the journal for references. However, some improvement is possible. For example:
Line 314: write “Salmonella” in italics.
Line 320: correct typo: “performance”, rather than “pwrformance”.
Line 365: remove “strmiska”.
Line 379: Do not write in italics. "L." (from Linneo in latin names of microorganisms).
To remove additional typos, I recommend revising the entire section.
TABLES:
Tables need to be interpreted independently of the manuscript text. Therefore, some improvement is needed:
Table 1: Remove horizontal line below "Dehydrated lucerne meal".
Table 1: Indicate in a footnote how ME value was obtained.
Table 2, title: Add: "(mean± SD)".
Table 2, heading: To avoid confusion with age, I recommend writing: "Day of fattening period", rather than “Day”
Table 2: revise footnote related to “a, b – mean values marked with different letters differ significantly at p ≤ 0.05” because this table does not display lettersarising from post hoc analysis.
Table 3, title: Add: "(mean± SD)".
Table 3, heading: To avoid confusion with age, I recommend writing: "Day of fattening period", rather than “Day”
Table 3, heading: Add heading row indicating "Significance of effects" (similarly to Table 2).

Author Response
Responses to Reviewers comments are in PDF file.

Reviewer 2 Report
The manuscript does not clearly state (abstract, discussion, conclusions) which treatment was the most suitable in terms of mineral composition and physicochemical properties of meat (treatments E, S, E+S). Related to that, the last sentence of the abstract seems to me a little too general.
Given the great variability (especially the mineral composition of meat), I believe that the research (related to meat analysis) needed to be conducted on a larger number of animals (only 6 animals per treatment) and then the differences between treatments would probably be more significant.
In addition, I do not recommend using manuscript title words for manuscript keywords.
Instead of term "time" (Tables 2 and 3) I suggest term "age".
Author Response

(The authors gave the same response as above.)
